# Construction of a Soundscape-Based Media Art Exhibition to Improve User Appreciation Experience by Using Deep Neural Networks

**Youngjun Kim, Hayoung Jeong, Jun-Dong Cho** and **Jitae Shin** *

College of Information and Communication Engineering, Sungkyunkwan University, Suwon 16419, Korea; yjk931004@skku.edu (Y.K.); lmo3088@skku.edu (H.J.); jdcho@skku.edu (J.-D.C.)
* Correspondence: jtshin@skku.edu

**Abstract:** The objective of this study was to improve user experience when appreciating visual artworks with soundscape music chosen by a deep neural network based on weakly supervised learning. We also propose a multi-faceted approach to measuring ambiguous concepts, such as the subjective fitness, implicit senses, immersion, and availability. We showed improvements in appreciation experience, such as the metaphorical and psychological transferability, time distortion, and cognitive absorption, with in-depth experiments involving 70 participants. Our test results were similar to those of "Bunker de Lumières: van Gogh", which is an immersive media artwork directed by Gianfranco Iannuzzi; the fitness scores of our system and "Bunker de Lumières: van Gogh" were 3.68/5 and 3.81/5, respectively. Moreover, the concordance of implicit senses between artworks and classical music was measured to be 0.88%, and the time distortion and cognitive absorption improved during the immersion. Finally, the proposed method obtained a subjective satisfaction score of 3.53/5 in the evaluation of its usability. Our proposed method can also help spread soundscape-based media art by supporting traditional soundscape design. Furthermore, we hope that our proposed method will help people with visual impairments to appreciate artworks through its application to a multi-modal media art guide platform.

**Keywords:** music recommendation system; multimedia data processing; weakly supervised learning; soundscape music; media art; exhibition environments; user experience

## 1. Introduction

A soundscape is a contextual acoustic environment perceived by humans. Initially, this concept was used as a compositional approach in order to provide a sense of space via the recording and rearrangement of noise, including natural and environmental elements. This role of the soundscape is used to help an audience imagine its space [1,2]. Recently, the concept of the soundscape has been used in a variety of fields, such as architecture, art, and education [3–6]. In particular, some research [7] has proposed art-based spatial design by presenting the concept of shopping galleries through a combination of shopping spaces and soundscape music. The concept of the soundscape in art education has also been explored [8]. Recently, IT technology [9,10] has been used to improve user experience in map applications through data-based soundscape construction methods.

In this study, we applied the soundscape concept to an art exhibition environment to improve the artwork appreciation experience. At this point, music is an important component of soundscape-based exhibition environments because it can interfere with or help an audience appreciate the experience. In previous studies, soundscapes were designed by experts by selecting or composing music containing a message that the experts wanted to convey. However, this approach is expensive in terms of time and effort. In this study, we investigated the replacement of this traditional approach to soundscape construction with deep-neural-network-based methods. Our requirement was that the

soundscape constructed using our method should provide as impactful of an experience as that of the expert's choice. If our requirements are met, our approach could spread the abundance soundscape-based media art at a lower cost than that which was previously possible.

The purpose of this study was to improve users' artwork appreciation experience through auditory cues, such as classical music, in order to provide an abundant media art exhibition environment. These multi-modal sense-based exhibits [11–14] not only provide the user with a more impressive, realistic, and immersive experience, but also have potential cognitive and emotional impacts on the appreciator [15–17]. Thus, we propose a soundscape-based methodology that uses deep neural networks to identify music associated with a given visual artwork through multi-modal data processing based on weakly supervised learning. With a multi-faceted approach, we measure whether our system can recommend music that can have an impact on the user. This soundscape-based media art will improve users' experience of appreciating artworks through metaphorical and psychological interactions as well as through the direct and material appreciation of media art. Our method of soundscape design using deep learning can help disseminate media art exhibitions by reinterpreting them as immersive media. Furthermore, we hope that our proposed method will help people with visual impairments to appreciate artworks through its application to a multi-modal media art platform. The contributions of this study are described below.

*Contributions*

1. We developed a system for matching classical music and paintings using the concept of the soundscape through an improved deep neural network based on weakly supervised learning.
2. We propose a multi-faceted approach to measuring ambiguous concepts, such as the subjective fitness, implicit senses, immersion, and availability; then, an interdisciplinary discussion is also provided.
3. We show the improvements in the appreciation experience, such as metaphorical and psychological transferability, time distortion, and cognitive absorption, with in-depth experiments involving 70 participants.

## 2. Related Studies

### 2.1. Multi-Modal Artwork Platform for People with Visual Impairments

Modern information systems rely on vision, resulting in differences in information gain between visually impaired people and non-visually impaired people. Technological developments have improved this situation [18,19], but access to cultural content beyond daily life remains challenging. In particular, persons with disabilities should have opportunities provided for cultural, physical, and artistic activities based on the 2008 Convention on the Rights of Persons with Disabilities [20]. Nevertheless, this has not been the case from the perspective of arts and culture, as pointed out by Kim Hyung-sik, a former member of the Committee on the Rights of Persons with Disabilities (CRPD) in the United Nations [21].

The blind touch project [22] was launched by ratifying Article 30 (2) of the Convention on the Rights of Persons with Disabilities, which states that "Parties shall take appropriate measures to enable persons with disabilities to have the opportunity to develop and utilize their creative, artistic, and intellectual potential, not only for their own benefit, but also for the enrichment of society". This project aimed to improve the artwork appreciation environment for people with visual impairments [23,24]. The blind touch project had two main objectives. The first objective was to allow blind people to experience, understand, and interpret art through various multi-modal senses, such as hearing, touch, temperature, and texture. The second objective was to develop a framework and technologies that would allow visually impaired people not only to experience art through their senses, but also to understand, interpret, and reflect upon it.

In a blind touch project, Cavazos et al. [25] developed a multi-modal artwork guide platform (see Table 1) that transformed an existing 2D visual artwork into a 2.5D (relief form) replica using 3D printing technology, making it accessible through touch, audio descriptions, and sound in order to provide a high level of user experience. Thus, visually impaired individuals could enjoy this artwork freely, independently, and comfortably through touch and sound without the need for a professional commentator. In addition, gamification concepts [26] were included to awaken various other non-visual senses and maximize enjoyment of an artwork. For example, vivid visual descriptions and sound effects were provided to maximize the sense of immersion in appreciating artworks. Such recreated artworks with multi-modal guides facilitated user-friendly interaction environments by sensing the event of tactile input on some part of an artwork and providing the related information. In addition, background music was created to elicit emotions similar to those of the work, taking into consideration the musical instrument's timbre, minor/major mode, tempo, and pitch. In this paper, we aimed to replace the background music with other classical music recommended by deep neural networks and soundscape concepts.

**Table 1.** Interactive multi-modal guideline for appreciating visual artworks and museum objects [25].

| Type | Description |
| --- | --- |
| Sensing technology | Capacitive sensor connected to conductive ink-based sensors embedded under the surface of the model. |
| Input | Double-tap and triple-tap gestures on the surface. |
| Tactile presentation | Tactile bas-relief model |
| Output | Audio descriptions, sound effects, and background music |
| Objective | Improve visual artwork exploration |

*2.2. Soundscape Construction Using Deep Neural Networks*

In this study, we constructed soundscapes based on music that matched well with a given artwork by using deep neural networks. We considered three technical approaches to constructing soundscapes. The first is a generative-model-based approach, in which generative model is used to translate a painting into music based on deep neural networks [27]. This proposed method is characterized by the use of consistent features that allow interconversion between music and painting. However, this method of using consistent and interchangeable features does not ensure well-matched music. Therefore, in this work, we did not adopt this method because we felt that it did not produce music that fits paintings well. Rather, we used the approach of finding music that matches the painting rather than using a generative models.

The second is an approach based on music recommendation systems. The recommendation systems were divided into user-based, content-based, and hybrid-based methods [28,29]. We focused on content-based recommendation systems [30] because our purpose was a recommendation between music and painting, not personalized recommendations. The key to this approach is the vectorization of content because it matches based on the similarities in vectorized content. Examples of vectorization methods include video and description summarization [31,32] and image captioning [33–35]. In particular, the authors of [36] matched poetry and images through captioning, the authors of [33] presented an automatic caption generation method for Impressionist artworks for people with visual impairments, and the authors of [28] used emotional features in music recommendations. However, when applying these methods to our research, not only was a capping module required for the music and images, but a user-based recommendation system was also needed.

Thus, we selected baseline networks as a third method [10,37–41] for feature matching by using kernel density estimation based on weakly supervised learning. Our training

baseline was soundnet [37], in which a kernel mapped an audio sample space to an image sample space via weakly supervised learning for vectorization. In prior imaginary soundscape research [10], an application based on soundnet was used to improve users' experience with Google Maps. This application mapped vectors of images and audio into the same sample space by using soundnet. Our approach uses a similar framework. This approach has the limitation of not being able to use audio or image descriptions, but it has the advantage of being able to focus on natural features. In the following sections, we describe our audio feature extraction method and knowledge distillation method.

### 2.3. Audio Feature Extraction

Audio feature extraction is a major component of soundnet frameworks. There are three major methods. The first deals with audio data as a spectrogram image by using a short-term Fourier transform [42–45]. The second is an end-to-end method for dealing with raw audio data by using a shallow and wide raw feature extractor [46,47] rather than a Fourier transform. The third one uses improved learning techniques, such as data argumentation [47–49], pre-processing (or post-processing) [50,51], and other learning methods, such self (or weakly) supervised learning [10,37–41,47,51].

Spectrogram-based audio feature extraction depends on the hyper-parameters of the short-term Fourier transform. Thus, many state-of-the-art networks [37,47,51] have been studied by using raw feature extractors; however, we adopted a spectrogram-based approach because it has advantages in terms of knowledge distillation and its application to our domain. In particular, we selected the WaveMsNet [45] as our baseline feature extractor. The WaveMsNet fuses features of time and frequency domains to improve the dependence of the Fourier transformation on the hyper-parameter window size via multi-scale feature extraction in the time domain. However, the network still receives the gray channel as the input. In this paper, we propose a multi-time-scale transform [52,53] to convert audio data into an RGB image in order to receive RGB input instead of gray input. These techniques not only improve the receptive field, but also enable direct measurement of feature distance. In the next sections, we shall address knowledge distillation for mapping audio features and image features into the same sample space.

### 2.4. Knowledge Distillation

Knowledge distillation began from mimic model [54], and is a method to learn differences among distributions for model compression. This method has therefore been used in various fields related to distribution learning, such as domain adaptation [55] and knowledge transfer [56]. This method is also used in soundnet frameworks as a learning method. There are two main considerations when applying knowledge distillation. The first is the choice of the knowledge to learn, such as score maps [57], feature maps [58], attention maps [59], Jacobian matrixes [60], or decision boundaries [61]. The second is how the distilled knowledge is transferred, such as through mutual learning [62], knowledge projection [63], or teacher assistance [64].

In this study, we selected two research studies as baseline studies. The first was that of fitnet [58], which conducted knowledge distillation based on a feature map with abundant information. A previous fitnet study solved the problem of size mismatch between feature maps via a regressor, which problem is to transfer of knowledge from wide features to narrow features Later, an attention study [59] showed that learning without an additional regressor was possible by changing the learning structure from deep to shallow. In this paper, we conducted knowledge transfer through direct measurement of feature distance without an additional regressor by setting the same model structure for transferring from wide and deep features to wide and deep features. Our learning method is based on a deep mutual learning strategy [62] with symmetric Kullback–Leibler divergence. Furthermore, the strategy used by the learning method allows the audio feature extractors to be configured in the same way as the image feature extractors. We

chose this learning method because it aims to reduce the difference between the two sample spaces rather than to increase the accuracy of specific tasks.

## 3. Proposed Architecture and Learning Method for Constructing a Soundscape

In this section, we deal with the proposed architecture of our application and present a learning method for constructing a soundscape in four subsections—namely, music–artwork matching for the soundscape, a training phase, a domain adaptation phase, and a multi-time-scale transform for audio feature extraction.

### 3.1. Music–Artwork Matching for the Soundscape

Figure 1 shows the architecture of our service application. In the preprocessing phase, music is converted into RGB images via a multi-time-scale transform. Multi-time-scale transform and audio feature extraction are performed, and the resulting data are stored in a database. Later, when a painting is entered through a service application, the application matches and recommends the nearest $n$ music pieces that match well with the audio features stored in the database. The distance is measured by the cross-entropy. A feature extractor for the audio and images was constructed by using a wide resnet 101 with double width. For audio with a standard sample rate (44.1 kHz), the extracted features are stored as a JSON object. The stored JSON object had about 3.48 times more capacity than the existing audio because of the feature characteristics, such as the high resolution and multiple channels. We did not use any additional compression methods. Our music database consisted of 2000 items of classical music stored in the form of key values, and it took about 2.5 days for the database to be configured without parallel processing. In addition, a total inspection was conducted via cross-entropy without an additional search algorithm for the music–painting matching, which took about 3.2 h. Our device settings were as follows: CPU: Intel® Core™ i7-8700K processor, GPU: 2 RTX 2080 Ti. However, because we used unoptimized code, the technical issues described above are likely to be improved in real-world service via optimization. In the next section, we describe how the deep neural networks were trained.

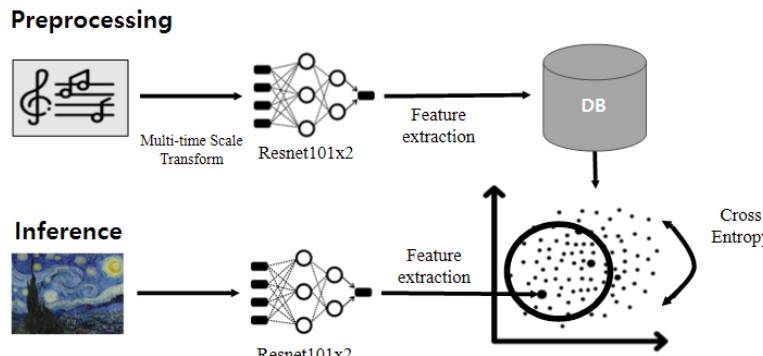

**Figure 1.** Our service application architecture, consisting of a preprocessing phase before the service and an inference phase at runtime. This figure shows a music–artwork matching method for soundscape construction.

### 3.2. Training Phase

Figure 2 shows our learning framework, which is similar to the soundnet framework. However, we additionally used methods such as a pre-trained model with shared features, an improved audio feature extractor, and improved weakly supervised learning. In the pre-training phase, a multi-domain convolutional neural network (CNN) was applied for sharing features. In the original soundnet framework, the feature extractor is based on two models—namely, the object and scene of the feature extractor for each pre-training phase and weakly supervised learning phase. However, the data cannot be utilized efficiently. In soundnet, the object feature extractor is learned in Imagenet, while the scene feature

extractor is learned in Place365, which results in less data than Imagenet. Thus, it is difficult to determine if the feature extractor has learned enough. To address this issue, we applied a multi-domain CNN because the place and object can share features when using this method. The feature extractor used was the wide resnet101 with double width and with two classifier headers for the object and scene. The Imagenet and Place365 data were trained in one model so that the features could be shared. This method also had advantages in the training phase.

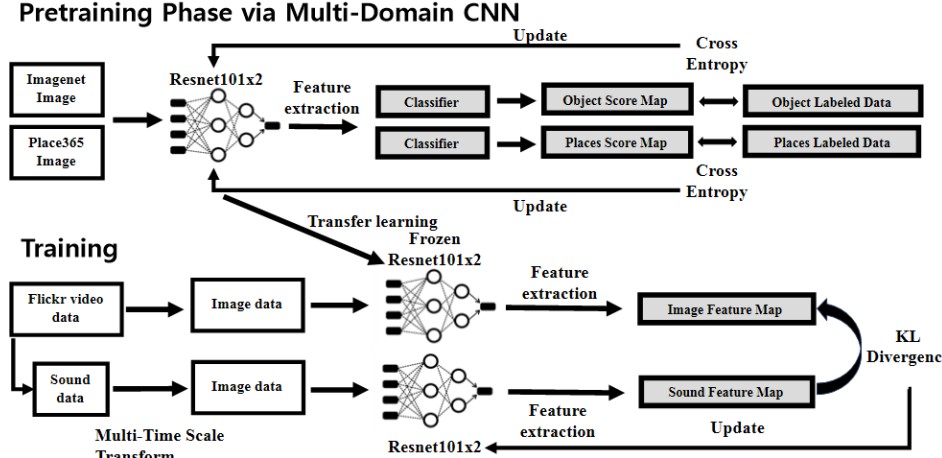

**Figure 2.** The training phase was divided into a pre-training phase and a training phase. This learning framework is similar to that of the soundnet framework; however, the pre-trained model with features shared via a multi-domain CNN, audio feature extraction via multi-time-scale transform, and weakly supervised learning via mutual learning are advantages of our framework.

In this training phase, we trained our network by using a Flickr video dataset [37] for cross-modal recognition. The video dataset was viewed as weakly labeled data with images assigned to the audio. Image features were extracted from a pre-trained model that was frozen. This frozen model meant that the results of learning were not reflected; they were used only for feature extraction. Audio data were then converted into RGB images via the multi-time-scale transform, which allowed feature extraction via the wide resnet with double width. These extracted features were used as the sources and targets for audio features and image features, respectively. The deep neural network was trained with a kernel function to map the source to a target, which meant that learning to extract features was similar for the audio and image data. Only the audio feature extractor reflected the learning results because our purpose was to approximate from the source to the target. Unlike soundnet, in the KL divergence, the source and target were configured as feature maps, not score maps. This is the same method as that used in the fitnet; however, our training frameworks conducted direct distribution training without an additional regressor through use of the same feature extractor structure. Furthermore, because the multi-domain CNN was applied in the pre-trained model, we also performed inter-distribution learning on one integrated model through only feature sharing.

*3.3. Domain Adaptation*

In the training phase, a network was trained to match audio with objects and scenes. In the domain adaptation phase, we developed a method to train our network to match music and a painting, which is challenging because of the absence of a related dataset. We therefore focused on appealing advertisements in our dataset. The purpose of emotionally appealing advertisements is to transfer emotional feelings to the customer rather than rational information. A new paradigm of emotionally appealing product advertisements has recently emerged, where the concept is not to focus on revealing production, but to convey a brand image and overall atmosphere. These advertisements convey a sense of

artistry and atmosphere that transcend the boundaries of usability and beauty by using well-matched colors and atmospheric music. Thus, the video of an advertisement can be considered as an expert's well-matched labeling data. In this work, we matched paintings and music by using the color and the audio atmosphere rather than simply matching audio to objects or scenes. Therefore, we used advertisements for products. Examples of the emotionally appealing advertisements used are provided in Figure 3. In this work, data were collected manually. The collection criteria were as follows: The first was whether background music was included; the second was whether the advertisements emphasized color; the third was whether there was conversation. If conversations were frequent, the advertising was excluded from the collection. If several atmospheric parts existed in the collected advertisement, the video was divided into several videos according to the sections where the atmosphere changed. For example, in Figure 3, each row is from the same advertisement. This advertisement was divided based on the points at which the atmosphere changed.

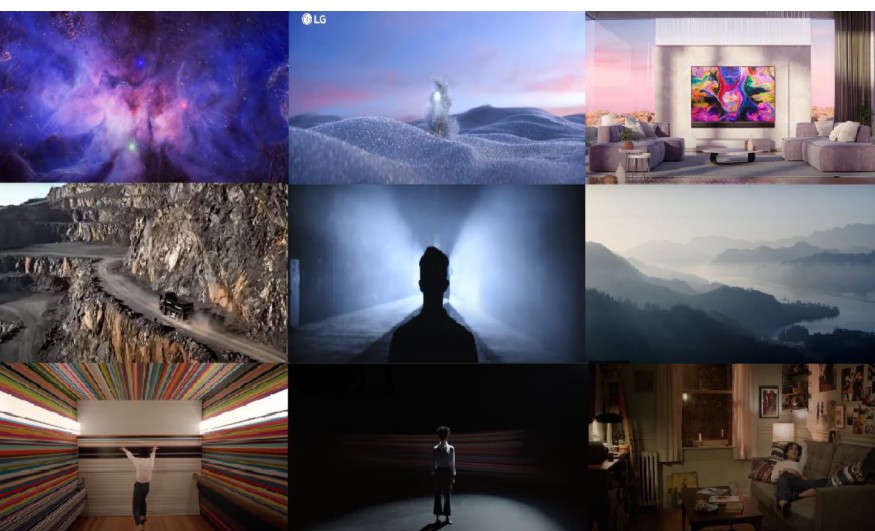

**Figure 3.** Dataset of emotionally appealing advertisements for domain adaption.

Figure 4 shows our domain adaptation method with its key features of mutual learning through advertisement data with symmetric *KL* divergence. The purpose of mutual learning was not to increase the accuracy of the task, but to train the two features to be the same. Therefore, we used advertising videos to fine-tune the network through mutual learning with symmetric *KL* divergence for domain adaptation. This domain adaptation had two main objectives. First, we wanted our system to learn a method of matching sounds and images based on considerations of the color and the atmosphere. Second, we focused on extracting these two features equally, rather than matching the sound to the space of the image via symmetric *KL* divergence.

$$D_{SymmetricKL}(P, Q) = D_{KL}(P \parallel Q) + D_{KL}(Q \parallel P) = \sum_{x \in \chi} P(x) \ln \frac{P(x)}{Q(x)} + \sum_{x \in \chi} Q(x) \ln \frac{Q(x)}{P(x)}. \quad (1)$$

Equation (1) is the symmetric *KL* divergence for the mutual learning used in this study. The objective of mutual learning is to train model so that image and audio features can be extracted equally; the image is indicated by *P* and the audio by *Q*. Importantly, unlike in the training phase, a model freeze was not used in order to reduce the gap between the distributions of the two features.

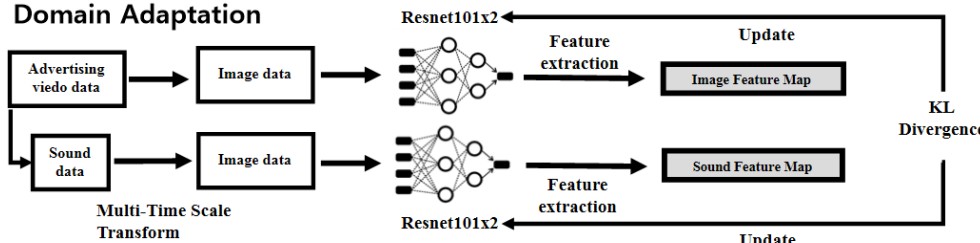

**Figure 4.** Domain adaptation phase with symmetric KL divergence based on mutual learning.

### 3.4. Audio Feature Extraction via Multi-Time-Scale Transform

Algorithm 1 is for audio feature extraction via the multi-time-scale transform in order to extract features from an RGB image rather than a conventional gray image. This method is an improvement over WaveMsNet [45,46], the Specaugment method [65], and time-wise multi-inference strategies. In the initialization section, the hyper-parameters of the FFT were experimentally obtained based on values that are commonly used in ESC-50, and the hyper-parameters of the MTST were obtained via a greedy search. The ranges of the greedy searches were as follows: The steps were [50, 100, 150, 200, 250, 300] and x_size was [224, 401, 501, 601, 701, 801]. M was a method of conversion from raw data into a mel-spectrogram with the conversion of power to decibels (dB). Audio features were then extracted in a similar manner, except that we used a multi-time-scale transform. The model that we used was the wide resnet101 with double width.

Figure 5 shows the key idea of the multi-time-scale transform. The two figures are the same graph, with the three-dimensional visualization shown on the left and the two-dimensional visualization on the right. The x-axis shows frequency, the y-axis shows time, and the *z*-axis shows power. A heatmap-based RGB image is shown on the left, while the multi-time-scale transform image is shown on the right. The heatmap-based RGB image was determined by the power. In other words, even when information from the three channels was combined, only the gray information was available because the gray information quantity was distributed among the three channels according to the power level. We distributed information into the RGB channel to increase the total information. The spectrogram was up-scaled to match input shapes via bilinear interpolation, then divided into three parts based on the time axis using multi-scale inference (see Algorithm 1).

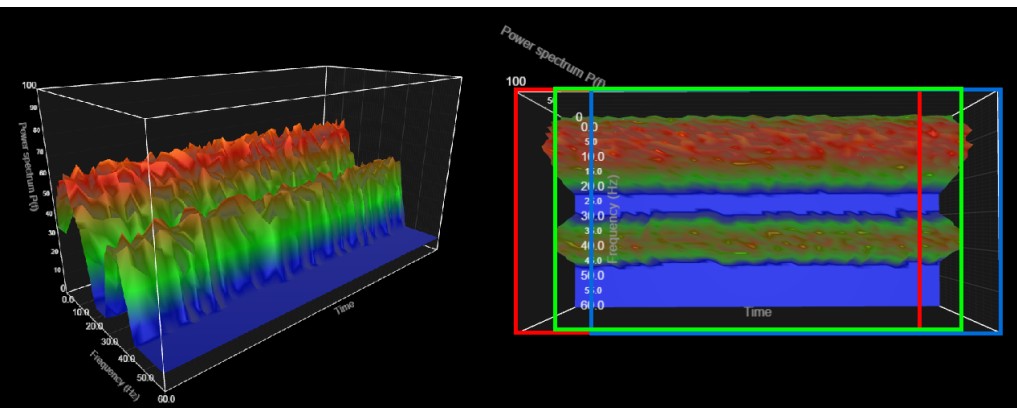

**Figure 5.** Lighting-chart-based simulation plots to help understand the key idea of the multi-time-scale transform.

---

**Algorithm 1** Audio Feature Extraction via the Multi-Time-Scale Transform

---

*Initialization*;
[x]←*audio_signal*, [t]←*time*, [n_fft]←4096, [hop_length]←441, [n_mels]←224,
 [f_min]←20, [f_max]←20000, [top_db]←160, [x_size] ←701, [y_size] ←224, [step]
 ←200;
**Audio Feature Extraction:**
    spectrogram = |M(x,n_fft,hop_length,n_mels,f_min,f_max,top_db)|;
    $image_{rgb} = MTST(spectrogram, size = (x\_size, y\_size, step), p = 0.5)$
    features = model($image_{rgb}$)
    **return** features;
**MTST** *(spectrogram, size, p)***:**
    x_size, y_size, step = size
    spectrogram = normalization(spectrogram)
    $image_{gray} = bilinear\_interpolation(spectrogram, (x\_size, y\_size))$
    r=spec_aug($image_{gray}[;, 0 : max\_size - step], p$)
    g=$image_{gray}[;, step/2 : max\_size - step/2]$
    b=spec_aug($image_{gray}[;, step : max\_size], p$)
    $image_{rgb} = concatenate(r, g, b)$
    **return** $image_{rgb}$

---

## 4. Experiments

We performed three main experiments. The first experiment was a quantitative evaluation in order to determine how effective the multi-time-scale transform was and if it improved the task of audio feature extraction. In the second experiment, we evaluated how reasonably music and paintings were matched by our system. In the third experiment, we evaluated whether the user experience improved while appreciating artworks and the soundscape music.

**Hypothesis 1.** *"If the soundscape music and painting are well matched, user appreciation of the experience will increase."*

The above hypothesis is the central focus of this study. However, assessing the match between the music and painting and the improvement of the user experience is challenging. We therefore developed strategies for measuring these factors, as described in the following subsections.

### 4.1. Audio Feature Extraction via the Multi-Time-Scale Transform

We performed an ablation and benchmark study to determine the effectiveness of the multi-time-scale transform. This experiment was conducted using the ESC-50 as an environmental sound classification dataset. The environmental settings were as follows: Pytorch 1.1 with Python 3.6. The batch size was 32, and a random shuffle was used. The optimizer used was Adam, and the initial learning rate was $1 \times 10^{-4}$; the scheduler used the cosine annealing learning rate scheduler. The hyper-parameter T value was 50. The hyper-parameters of training were obtained via a greedy search. The ranges of the greedy searches were as follows: The batch sizes were [8, 16, 32], and the initial learning rates were $[1 \times 10^{-4}, 2 \times 10^{-4}, \ldots, 5 \times 10^{-4}, 1 \times 10^{-5}, 2 \times 10^{-5} \ldots 5 \times 10^{-5}]$.

Table 2 shows the results of the ablation study. First, we conducted experiments using various backbone network settings, such as Mobilenet v2, Efficientnet, VGG, Resnet, and Densenet, with different widths and depths. However, networks other than Resnet and Densenet were omitted from Table 2 due to their poor results. We increased the depth of the network-provided setting and the width by one, two, and four times the depth. The width showed the best performance when it was doubled, and the depth tended to be better, but not in all cases. Resnet101 with double width showed the best performance. In

Table 2, the application of our multi-time-scale transform to resnet resulted in about 2.8% higher accuracy than that of the baseline network, WaveMsNet.

**Table 2.** Ablation study of the multi-time-scale transform.

| Methods | Accuracy |
|---|---|
| Densenet with spectrogram | 73.9% |
| Resnet with spectrogram | 76.3% |
| WaveMsNet after first phase [45] | 70.05% |
| WaveMsNet after second phase [45] | 79.1% |
| Resnet with multi-time-scale transform | 81.9% |

Table 3 shows the sound classification results for various methods based on the ESC-50 benchmark. Methods without an asterisk in Table 3 are examples of supervised learning methods that use backbone and data argumentation methods, while methods with an asterisk were trained using various methods, such as weakly (or self) supervised learning, between-class learning, or some other method. Our network showed better performance than that of baseline, but lower performance than that of the state-of-the-art methods. AclNet uses raw feature extractors, such as Envnet v2. This is a weakness for mutual learning; therefore, we did not use these methods. Ensemble-fusing CNN is a state-of-the-art method, but is not a single-model method. Our model performed as well as the single-model methods, and had advantages for mutual learning because it is spectrogram-based. Therefore, in this work, we selected our network as the baseline. However, in our future research, various augmentations with ensemble-fusing CNNs should be explored for applications in our work. Our model's performance was about 12.2% poorer than that of WEANET. Therefore, work is needed to enhance the performance of our network. Nevertheless, our network has the advantage of being spectrogram-based and showed better performance than human performance. The validation results for each fold of our proposed method can be found in detail in Table 4.

**Table 3.** ESC-50 dataset benchmark.

| Methods | Accuracy |
|---|---|
| EnvNet v2 + S.A. [47] | 78.8% |
| WabeMsNet (our baseline) [45] | 79.1% |
| Resnet with Multi-Time-Scale Transform | 81.9% |
| AclNet [49] | 85.65% |
| Ensemble-fusing CNN [48] | 88.65% |
| Soundnet (our baseline) [37] * | 74.2% |
| EnvNet v2 + S.A. + B.C.L. [47] * | 83.9% |
| WEANET [66] * | 94.1% |
| Human Performance [67] | 81.3% |

### 4.2. Relevance of Music–Artwork Matching

**Definition.** ***"Well-Matched Soundscape Music–Painting"***: In this work, we defined details factor of "Well-Matched Soundscape Music–Painting" for a measurement of the quality of a match. Blind touch is intended to provide direct and material interactions that go beyond metaphorical and psychological interactions between the art and the appreciator. However, this does not imply the absence of traditional art. Therefore, in this study, we wanted to measure the metaphorical and psychological relevance of matching between music and a painting. The metaphorical scale measured how much the same implicit multi-sensory experience was transferred because blind touch is a multi-sensory-based media art. The psychological scale was measured through subjective fitness, which is

an individual's appreciation, because it is the most representative single scalar value of individual appreciation.

**Table 4.** Result of five-fold cross-validation. The F1 score is the weighted F1 score and the MCC is the Matthews correlation coefficient, which was obtained as the average of the MCC for each class. This metric tables were written by referring to [68,69].

| Fold | Accuracy | Recall | Precision | F1 | MCC | AUC |
|------|----------|--------|-----------|------|------|------|
| 1 | 81.2% | 81.2% | 84.4% | 81.1% | 81.0% | 90.4% |
| 2 | 81.0% | 81.0% | 83.5% | 80.4% | 80.7% | 90.3% |
| 3 | 81.2% | 81.2% | 82.6% | 80.7% | 80.9% | 90.4% |
| 4 | 84.8% | 84.8% | 85.7% | 84.2% | 83.7% | 91.8% |
| 5 | 81.2% | 81.2% | 84.0% | 81.1% | 80.9% | 90.4% |

**Experimental Setting:** Twenty-four men and 16 women, all of whom are Korean, were evaluated in this study. Each group consisted of 10 people divided into four groups with a balanced gender ratio. The criteria for dividing the groups were as follows. First, the artistic culture of the subjects was evaluated via a simple test with the music and paintings used in this experiment. Second, the subjects were allocated to each group based on their test scores, which we averaged. The average music score was 2.3, the average painting score was 3.4, and a perfect score was 5. The roles of each group was as follows: Group 1 participated in subjective fitness experiments. Groups 2 and 3 participated in multi-sensory concordance experiments. Group 4 participated in multi-sensory concordance inverse measurement experiments. The groups were designed to avoid duplication of the experiments and to ensure that prior knowledge did not affect the experimental results. Furthermore, we did not provide any information other than information related to the experimental progress. This meant that the subjects did not know that the paintings and music were recommended via a deep neural network. Figure 6 shows the paintings used in our experiment, and the music allocated to the paintings by our system is indicated in Table 5.

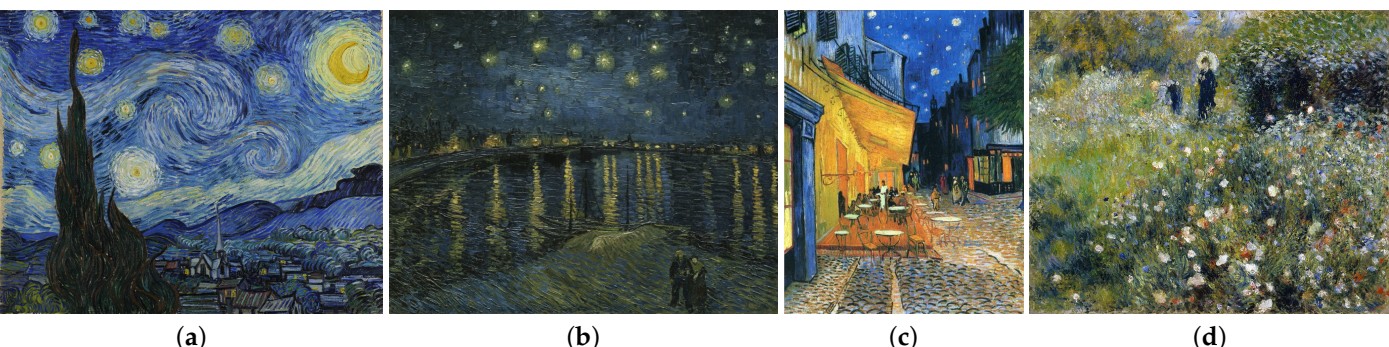

(a)  (b)  (c)  (d)

**Figure 6.** Paintings used in the experiments. (**a**) Starry Night; (**b**) Starry Night over the Rhône; (**c**) Café Terrace at Night; (**d**) Femme avec parasol dans un jardin. This figure is associated with Table 5.

**Table 5.** Music–painting matching scheme. The top three pieces of music selected by our system for each painting are shown. Rows 1, 2, and 3 correspond to musical choices 1, 2, and 3, respectively.

| Painting | Music |
|---|---|
| (a) | Beethoven: Piano Sonata No. 3 in C, Op. 2<br>Prokofiev: 10 Pieces from Romeo and Juliet, Op. 75, No. 9, Dance of the Girls With Lilies<br>Grofé: Grand Canyon Suite—4. Sunset |
| (b) | Edvard Grieg—Peer Gynt—Suite No. 1, Op. 46—III. Anitra's Dance<br>Bolero—Maurice Ravel<br>Ashkenazy: Sibelius—Valse Triste, Op. 44 |
| (c) | Tchaikovsky—The Sleeping Beauty Suite, Op. 66a<br>Joseph Haydn—Symphony No. 88 in Major, Largo<br>Verdi: "Nabucco"—Schippers |
| (d) | Jean Sibelius—Valse Triste, Op. 44, No. 1<br>Edvard Grieg: "Peer Gynt—Morning Mood"<br>Tchaikovsky—Swan Lake Op. 20, Act III |

### 4.2.1. Measurements of Subjective Fitness

**Experimental Methods:** Subjective fitness was measured in group 1, which comprised five men and five women. First, we input the paintings into our system and extracted the top three music recommendations. Second, when both the paintings and music were experienced at the same time, we measured the subjective fitness using a five-point Likert scale. We changed the paintings in the painting–music pairs to ensure that previous and subsequent experiments did not affect the current experiments; for example, painting 1–music 1, painting 2–music 1, painting 3–music 1, painting 4–music 1, then painting 1–music 2, painting 2–music 2, etc.

Table 6 shows the results of the subjective fitness experiments. Columns (a), (b), (c), and (d) are associated with Figure 6, Music pieces 1, 2, and 3 are associated with Table 5, and Music 4 was used in the soundscape music exhibition [12] "Bunker de Lumières Van Gogh" held in Jeju, Korea. Therefore, Music 4 can be considered to be a type of ground truth labeled by experts. F-score is the fitness score, and the P-score is the preference ratio. The fitness score was measured on a five-point Likert scale, and the P-score was the proportion of people who scored three or more F-score points. Music 4 used the same values as in previous studies [70]. Therefore, the system matched the artwork and music well in terms of subjective fitness. However, the P-score was not stable for Music 2, and Music 3 had a low F-score and P-score. The average F-score for all music items was 3.24, and the average P-score for all music items was 75%. The average F-score of Music 1 was 3.68, and the average P-score of Music 1 was 97.5%. This result is similar to that reported in a previous study, which reported an average F-score of 3.16, an average F-score for Music 1 of 3.74, an average P-score of 87.6%, and an average P-score of Music 1 of 94%. Thus, our system matched music and artworks well, but did not perform better than previous systems, despite the improved feature representation. This is likely due to the small size of the music database. However, the reliability of the system was demonstrated by the attainment of results similar to those reported in previous studies. In future studies, we need to assess whether the stability and performance of the system increase in response to the expansion of the music database.

**Table 6.** Results of the subjective fitness experiment. The F-score is the fitness score and the P-score is the preference ratio. Fitness was measured on a five-point Likert scale, whereas the P-score reflects the proportion of individuals with three or more points.

| Painting | Measure | Music 1 | Music 2 | Music 3 | Music 4 |
|---|---|---|---|---|---|
| (a) | F-score | **3.8** | 3.2 | 3.1 | 3.77 |
|     | P-score | **100**% | 80% | 60% | 85% |
| (b) | F-score | 3.8 | 3.6 | 3.2 | - |
|     | P-score | 100% | 80% | 70% | - |
| (c) | F-score | 3.4 | 3 | 2.7 | **3.85** |
|     | P-score | **90**% | 60% | 40% | 85% |
| (d) | F-score | 3.7 | 3 | 2.4 | - |
|     | P-score | 100% | 70% | 50% | - |

### 4.2.2. Measurements of Implicit Multi-Sensory Concordance

**Experimental Methods:** This experiment was conducted in groups 2 and 3, each consisting of five men and five women. The members of group 2 first wrote a review after viewing the four paintings. Group 3 then wrote a review after listening to the 12 music items. At this time, the subjects were not provided any information about the content of the experiment; the only guidelines that they received were to use all five senses when assessing the artworks or music. Fourth, we measured multi-sensory concordance by comparing the sensory language similarity of the reviews. We mapped words in the review using a Korean sensory word classification table (Table [71]). Sensory word tables are classified as gustatory, tactile, and temperature sensations; examples of such words are bitter, sweet, salty, sour, nutty, astringent, spicy, plain, rough, smooth, soft (texture), soft (material), hard, moist, sharp, cold, cool, lukewarm, warm, hot, etc.

Table 7 shows the results of the implicit multi-sensory concordance experiments. The D-score refers to the Euclidean distance and the C-score to cosine similarity. Table 7 shows that similar sensory trends between the two groups were measured through the C-score. However, it can be confirmed that the C-score is not proportional to Table 6's P-score or F-score because the cosine similarity is advantageous for distance measurements in high-dimensional positive space, but the size of each dimension is not meaningful. In other words, only the trends in each dimension can be evaluated, while the size of the difference is difficult to assess. We used the D-score to overcome this limitation. The C-score in Table 7 indicates the sensory similarity between both the music matched by experts and the music matched by our system. The D-score was similarly measured. The results show that not only can media art provide similar multi-sensory appreciation, but our system can also provide similar results to those provided by experts. However, this result has the limitation that a criterion was not provided to determine the significance according to magnitude of the score. Future studies should develop a criterion to determine the significance of the score's magnitude.

### 4.2.3. Inverse Measurements of Implicit Multi-Sensory Concordance for Validation

**Experimental Methods:** This experiment was conducted in group 4, which comprised nine men and one woman. The purpose of this experiment was to verify the experiments in Section 4.2.2. A questionnaire was created based on a multi-sensory word table constructed in Section 4.2.2. The experiment was conducted in the same order as that described in Section 4.2.1, but the questionnaire was completed instead of a written review. The questionnaire evaluated the extent to which the appreciator agreed with the sensory table using a five-point Likert scale.

Table 8 presents the results of the inverse implicit multi-sensory concordance experiment. The A-score is the agreement scale, which was the average score obtained using the five-point Likert scale. The C-score is the cosine similarity of the A-scores calculated based

on an active value (3 or higher). The purpose of this experiment was to measure how much the appreciator agreed with the implicit multi-sensory data. The A-scores and C-scores were low because of the failure of the experiment. Several problems were encountered during this experiment. The first problem was the mechanical marking phenomenon. When responding to the questionnaire, subjects gave low scores to senses that were mechanically opposed to the first high score. This was in contrast to the phenomenon where opposite senses were expressed together in the free reviews of appreciation. The second problem was the phenomenon of monotonous responses, which occurred when subjects became familiar with the experiment. This phenomenon demonstrated the tendency of subjects to exclude complex senses before appreciation. For example, the subjects gave low scores to combinations of options, such as the bitter taste of wine and the sour taste of candy, before even listening to music. Thus, in further studies, the validation of the experimental design should be improved.

**Table 7.** Results of the implicit multi-sensory concordance experiments. The D-score is the Euclidean distance and the C-score is the cosine similarity.

| Painting | Measure | (a) | (b) | (c) | (d) |
|----------|---------|-----|-----|-----|-----|
| Music 1 | D-score | 41.8 | 46.9 | 54.8 | 74.2 |
|         | C-score | 92% | 89% | 88% | 75% |
| Music 2 | D-score | 48.9 | 50.9 | 52.4 | 55.7 |
|         | C-score | 89% | 87% | 87% | 87% |
| Music 3 | D-score | 57.9 | 56.6 | 57.8 | 60.0 |
|         | C-score | 85% | 82% | 87% | 84% |
| Music 4 | D-score | 56.6 | - | 44.2 | - |
|         | C-score | 85% | - | 92% | - |

**Table 8.** Results of the inverse implicit multi-sensory concordance experiment. The A-score is the agreement scale and the C-score is the cosine similarity.

| Painting | Measure | (a) | (b) | (c) | (d) |
|----------|---------|-----|-----|-----|-----|
| Music 1 | A-score | 2.09 | 2.34 | 2.73 | 2.68 |
|         | C-score | 8% | 10% | 14% | 27% |
| Music 2 | A-score | 2.45 | 2.55 | 2.62 | 1.46 |
|         | C-score | 12% | 12% | 12% | 15% |
| Music 3 | A-score | 2.89 | 2.83 | 2.89 | 1.62 |
|         | C-score | 16% | 15% | 15% | 16% |
| Music 4 | A-score | 2.82 | - | 2.21 | - |
|         | C-score | 15% | - | 8% | - |

*4.3. Improvement of the Appreciation Experience with the Soundscape*

**Definition. "Improvement of the appreciation experience"**: In this work, we defined the measurable factor of "Improvement of the appreciation experience" as the combination of appreciation and subjective satisfaction; however, this is an ambiguous concept. To address this, we proposed an evaluation method for the immersion based on the flow theory of Csikszentmihalyi. We differentiated flow and cognitive immersion in this study according to flow theory. Flow was indirectly measured via time distortion phenomena, and cognitive absorption was measured through a simple test of working memory and attention concentration. The subjective satisfaction score of the appreciation experience comprised experiences of the environment and appreciation. The subjective environmental satisfaction was not related to improvements in the appreciation experience; however, the environmental score was a useful indicator of if the environmental setting was appropriate.

Appreciation scores were measured with a questionnaire based on the SSID [72] and WHO-5 Well-Being indices. The SSID is an evaluation index for soundscapes, and the WHO-5 index is an evaluation index for quality of life. Our questionnaire was prepared based on the SSID and WHO-5 Well-Being indices.

**Experimental Setting:** The participants in this experiment included a total of 29 men and 1 woman. They were divided into three groups, with each group consisting of 10 participants. The roles of each group were as follows: Groups 5 and 6 participated in the immersion experiments. Group 7 participated in the subjective satisfaction experiments. The criteria for dividing the groups and other experimental conditions were similar to those described for the previous experiments.

### 4.3.1. Measurements of Immersion

**Experimental Methods:** This experiment was conducted in groups 5 and 6. The purpose of this experiment was to indirectly measure immersion. Time distortion was measured for the flow measurement experiments. Group 5 appreciated artworks without listening to music, and group 6 appreciated artworks while listening to music. Time distortion was measured as the subjective assessment of time between the two groups, and flow was evaluated indirectly from the time distortion measurements. Cognitive absorption was measured with a simple test of working memory and attention concentration. This test asked questions about the color, location, shape, and texture of an object or scene.

Table 9 presents the measurement results for the immersion experiment. Each row presents the subjective times that participants felt while appreciating the exhibition. The ground truth refers to the real length of the piece of music. The *p*-value is the result of the *t*-Test, which was performed under the assumption that the variances were different. The results of the time distortion experiment indicate that group 5 predicted a time closer to the ground truth than group 6. In particular, participants in group 6 felt that more time had elapsed than actually had. Two interesting phenomena were discovered during the analysis of the interviews. The first was the "sleepy" phenomenon. When music pieces 1 or 4 were played, participants in group 5 did not experience sleepiness, while more than 80% of the participants in group 6 experienced sleepiness. The subjective viewing time was 2 m 36 s on average, while the experimental time was longer than the appropriate viewing time, ranging from 3 m 37 s to 6 m 52 s. The second phenomenon was the phenomenon of ambiguous answers. Group 5 answered with specific times, such as 4 m 20 s and 5 m 30 s, while group 6 tended to answer with ambiguous numbers, such as "about 5 minutes" and "about 10 minutes". In the experiments on cognitive absorption, the averages of the answers given by each group in our simple test were used. The perfect score for this test was 10. As shown in Table 8, groups 5 and 6 had an average score difference of 2.2. This can be attributed to the sleepy phenomenon. Thus, with the results obtained from this experiment, it was not possible to accurately confirm whether cognitive absorption was affected or not. However, music is qualitatively conducive to cognitive absorption. Examples of answers from participants included the following: **"The bouncy rhythm in the song reminded me of stars"; "The music felt like a young man in the country was leaving the village, and it helped me remember because there was a real village in the artwork"; "it wasn't hard to find the location of the lover because I watched the couple carefully because of the sentimental music being played."**

**Table 9.** Results of immersion experiments for assessing time distortion and cognitive absorption. A *t*-test was performed with the assumption of different variances.

| Painting | Music 1 | Music 2 | Music 3 | Music 4 | Avg. Answer |
|---|---|---|---|---|---|
| Group 5 | 8 m 18 s | 3 m 58 s | 3 m 32 s | 5 m 13 s | 6.4 |
| Group 6 | 10 m 22 s | 4 m 43 s | 5 m 18 s | 7 m 20 s | 4.2 |
| Ground Truth | 6 m 52 s | 3 m 37 s | 4 m 05 s | 5 m 33 s | - |
| *p*-value | 0.00137 | 0.00123 | $5.51 \times 10^{-7}$ | $1.03 \times 10^{-10}$ | 0.00511 |

### 4.3.2. Measurements of Subjective Satisfaction

**Experimental Methods:** This experiment was conducted in group 7. The purpose of this experiment was to measure the subjective satisfaction score. The subjects appreciated the paintings with the top music and filled out a questionnaire that was prepared based on the SSID and WHO-5 Well-Being indices.

The subjective satisfaction scores, which were based on the environment and the appreciation experience, are shown in Table 10. The top three questions were about environmental experience. The environmental satisfaction scores ranged from 3.3 to 3.6, which are fairly high scores. For the first question, discomfort was associated with 0 points and comfort with 5 points. Therefore, this experiment was appropriately constructed to assess the appreciation experience. The subjective satisfaction scores of appreciation ranged from 3.1 to 4.0. Therefore, soundscape music can aid in appreciation.

**Table 10.** Questionnaire based on the SSID and WHO-5 Well-Being indices.

| Question | Avg. Score |
| --- | --- |
| Are you feeling uncomfortable with your appreciation? | 3.3 |
| Is the volume of the soundscape music appropriate for you? | 3.5 |
| Is the sound quality of the soundscape music being played good? | 3.6 |
| Did the soundscape music go well with the painting? | 3.7 |
| Did music help you to enjoy paintings more? | 4.0 |
| Did music help you to be more comfortable when appreciating the paintings? | 3.1 |
| Did music help you to take initiative and actively appreciate the paintings? | 3.8 |
| Did music make your appreciation of the paintings fresher? | 3.3 |
| Did music make your appreciation of the paintings more interesting? | 3.3 |

### 5. Discussion

In this section, we would like to conduct an interdisciplinary analysis of how this metaphorical and psychological transfer could have occurred. This interdisciplinary approach consists of five parts: The first is the consideration of the inter-sensory transition phenomenon through the concept of synesthesia; the second is an artistic approach based on the characteristics of the paintings that we used in our experiments; the third is the consideration of the characteristics of deep neural networks in connection with the concept of synesthesia and the characteristics of our paintings; the fourth includes the limitations of this study and the future directions of development; the fifth is the overall blueprint of our research.

First, we deal with synesthesia by focusing on the transition of senses. References [15–17] show that media art can give potential cognitive and emotional impacts. We would like to discuss why this effect could occur. Synesthesia is a blending of the senses in which the stimulation of one modality simultaneously produces sensations in a different modality. Synesthesia is known to affect four percent of the population. However, via color–alphabet experiments with 400 people, studies [73] have shown that people without synesthesia perceive synesthesia but do not consciously recognize it. In other words, non-synesthesia implies that synesthesia is being perceived unconsciously. In particular, the study by [74] dealt with implicit associations between color and sound, particularly showing that various color properties can be mapped to acoustic properties, such as tone and volume. The study by [75] showed that the properties of sounds can be associated with colors. The study by [76] also demonstrated empirical investigations of the associations between auditory and visual perception. Based on these experimental results, transitions between implicit senses are possible; for example, value and hue can be mapped onto acoustic properties, such as pitch and loudness, pitch can be associated with color lightness, loudness can be mapped onto greater visual saliency, high loudness can be associated with orange/yellow

rather than blue, and chroma can have a relationship with sound intensity. The following sections cover the features of our artworks.

Second, the artworks used in our experiments were Impressionistic paintings. Impressionism is a trend in art that focuses on colors, lighting, and textures, and it is characterized by the ability to describe nature in the changing colors of light and to accurately and objectively record the visible world using the momentary effects of colors and shades. In particular, three of our four paintings were by Van Gogh, and he is known as a post-impressionist. Van Gogh tried to thicken paint through the Impasto technique to express the maximum texture and create a three-dimensional effect. In addition, Van Gogh did not describe objects in the same way, but captured his emotion and feeling very strongly through the touch of his brush. This distorted form, intense color, and simplification of the form were sought to express emotions. The other painting was "Woman with a Parasol in a Garden". Renoir was an early impressionist who understood how to express the effects of light, and was known to use black to express shadows. Therefore, the paintings we used are from the early to the late Impressionist period, and they are characterized by their good representation of colors, light, and texture. Therefore, we selected our paintings in consideration of these features. The characteristics of the paintings were analyzed and the possibilities of sensory transitions were addressed.

Third, we discuss how a deep neural network was able to recommend music and paintings in the above consideration. The study by [77] dealt with the perspectives of beauty, sentiment, and remembrance of art in deep neural networks. The study by [77] quantitatively and qualitatively analyzed three subjective aspects of human consciousness: image features in relation to aesthetics, sentiment, and memorability. This study indicated that the CNN considered features of various aspects, such as color, intensity, harmony of colors, object emphasis, pattern, art style, semantics, genre, and content. This study also addressed how deep neural networks judge and remember emotional and artistic values by exploring predicted aesthetic and emotional memory scores in the context of art history. In particular, these experiments also showed that aesthetic and emotional scores and color correlations are consistent with common assumptions. These experimental results show that deep neural networks consider factors that can cause sensory transference. Therefore, these features make our sensory transition possible.

Fourth, we address the limitations of our experiment and the directions of future study. There is a limitation in that no global solution was shown. The limitations can be seen from three perspectives: The first is that our research only dealt with Impressionist paintings from an artistic point of view. Therefore, further study is needed as to whether such studies can be applied to different artistic styles beyond Impressionism. The second is the problem of generalization; this study is not representative of the general population because the experiments were conducted on people in their 20s in Korea. In addition, research should be conducted on whether the experimental results are universal or if they are due to social learning. If the results obtained are due to social learning, extended studies should also be carried out from various perspectives, including culture, age, and local areas. Future research will therefore need to extend these local solutions into a global solution. The third is the effectiveness of media art, which can have potential cognitive and emotional impacts. However, our research focused on the potential cognitive impacts of the experiments. Therefore, we measured emotional impact with the F-score, the most representative and implicit single scalar value. These representative features of F-score can be influenced by various factors, such as emotions and environments, as well as by potential cognitive factors. Therefore, more detailed measurements should be studied in the future.

Fifth, our experiment was intended to enhance the experience of appreciation of media art through the effect of visual and auditory transitions. The transitions of these senses can be expected not only in vision and hearing, but also between various other senses, such as vision and smell or vision and touch. In this study, we aimed to create a platform that guides blind people through sensory transition they appreciate media art. Finally, we hope

that this multi-sensory experience will be harmonized and become abundant in media art. Furthermore, we hope that our proposed method will be applied to multi-modal media art platforms to help visually impaired people appreciate artworks.

## 6. Conclusions

This study described the construction of a soundscape-based exhibition environment using deep neural networks. The baseline was improved by different perspectives, such as modeling methods, learning methods, and domain adaptation methods. In addition, the soundscape music selected by our system was output via hyper-orientated speakers to improve the appreciation experience. To measure the improvements in user experience, we devised a soundscape music evaluation method and an appreciation experience evaluation method and conducted extensive experiments with 70 subjects. However, our research suffered from three major limitations. First, this was not state-of-the-art performance from the perspective of the deep neural network. In particular, models with low accuracy from the feature extraction perspective were selected for mutual learning. Therefore, the development of models with improved audio feature representation is required. Second, our current database comprised only 2000 music items. In addition, these 2000 music pieces were limited from the genre perspective. A wide spectrum of databases containing music from various cultures and times should be used in future studies. Third, we conducted experiments using 70 individuals, but each experiment was conducted on a group of only 10 people. Therefore, our experimental results cannot be generalized. Large-scale experiments need to be conducted to generalize these experimental results. This study shows the results of a pilot test with sighted test participants. We hope that the results of this study help people with visual impairments to appreciate art and that they help promote the cultural enjoyment rights of people with visual impairments.

**Author Contributions:** Conceptualization, methodology, and software, Y.K.; Data collection, experiments, and data curation, H.J.; Review and validation of UX experiments, J.-D.C.; Writing—review and editing, funding acquisition, and project administration, J.S. All authors have read and agreed to the published version of the manuscript.

**Funding:** This research was supported by the MSIT (Ministry of Science and ICT), Korea, under the ITRC (Information Technology Research Center) support program (IITP-2021-2018-0-01798) supervised by the IITP (Institute for Information and Communications Technology Promotion).

**Informed Consent Statement:** Informed consent was obtained from all subjects involved in the study.

**Conflicts of Interest:** The authors declare no conflict of interest. The funders had no role in the design of the study; in the collection, analyses, or interpretation of data; in the writing of the manuscript, or in the decision to publish the results.

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
