# Peer review of "Construction of a Soundscape-Based Media Art Exhibition to Improve User Appreciation Experience by Using Deep Neural Networks"

_electronics, doi:10.3390/electronics10101170_

Round 1
Reviewer 1 Report
This study proposed deep neural network architecture for user appreciation experience from soundscape-based media art exhibitions. The idea may be reasonable, but there are some major points that need to be addressed:
1. How did the authors deal with hyperparameter optimization of the models?
2. It is unclear on the datasets that the authors use in their experiments. The authors should describe this part clearly.
3. In some experiments, the authors only concerned with one metric (i.e., accuracy), it is important to report some more metrics.
4. Source codes should be provided for replicating the methods.
5. In the comparison, did all studies use the same evaluation method?
6. When comparing the predictive performance among different methods/studies, the authors should perform some statistical tests to see the significant differences.
7. Deep neural network has been used in previous works i.e., PMID: 33260643 and PMID: 31750297. Therefore, the authors are suggested to refer to more works in this description.
8. Is there any validation data on the experiments?
9. There must have space before any reference number.
Author Response
Please refer the attached file for our response.

Reviewer 2 Report
The authors designed a study for building a Sounscape-based Meda Art Exihibition with the aim to improve User Apprciation Experience using Deep Neural Networks.
The authors should be better described the novelties of their study with respect to existing ones. In particular, the author should discuss limitation and cons of the examined approaches. Furthermore, the authors should provide more details and discussion about the obtained results. In particular, the authors should provide more details about the used dataset, also providing a dataset characterization. Furthermore, there are any discussion that the authors can made about the preferences on the basis of gender or age?
I suggest to further analyze more recent approaches about the examined topics. In particular, I suggest the following papers to investigate emotional state of users by using deep learning techniques and the analysis of multimedia objects in the introduction section:
1) An emotional recommender system for music. IEEE Intelligent Systems.
2) Efficient music recommender system using context graph and particle swarm. Multimedia Tools and Applications, 77(2), 2673-2687.
Finally, I suggest to perform a linguistic revision.
Author Response

(The authors gave the same response as above.)

Reviewer 3 Report
In this article, the authors propose a method to construct an exhibition environment for blind touch, a tactile multimedia art form, through soundscape music chosen using deep neural networks and weakly supervised learning.
The paper is well organized, with proper structure, readability and length. The bibliography is sufficient and well given.
Specifically, the technical terms have been explained detailly and, the topic of this article is clear and understandable.
The presented methodology and the results are communicated clearly. The necessary background for the readers included in the paper.
The review of the state-of-the-art is sufficient. It includes lots of references to other relevant studies that have been previously proposed for the discovery of relations. Also, the novel contribution of the paper is highlighted, as well.
There is a clear presentation of the results and their commentary. The authors have made a coherent, accurate and focused presentation.
In general, the article has a very well-formulated problem and original propositions for its solutions.
Author Response
We want to give our sincere thank you for the summary of our paper and our contribution. It is very encouraging comments to us. Thank you again.
Round 2
Reviewer 1 Report
Most of my previous comments have been addressed. However, there are some points that still need room for improvement.
1. According to my previous comments, more measurement metrics should be added. It is not for comparison only, it also aims to evaluate whether the model worked well or not. Because the only accuracy can't be enough to consider a good model.
2. Measurement metrics (i.e., accuracy) have been used in previous machine learning-based studies i.e., PMID: 32942564 and PMID: 33260643. Thus the authors should cite more works in this description.
3. Releasing the source codes is indeed an important step for reproducing the study.
4. To evaluate the predictive performance, the authors should have some validation data.
Author Response

(The authors gave the same response as above.)

Reviewer 2 Report
I think that the authors have addressed all my concerns.
Author Response
Thank you for agreeing our response.
Round 3
Reviewer 1 Report
My previous comments have been addressed well.